# The effects of COVID-19 mitigation measures on physical activity (PA) participation among adults in Rwanda: An online cross-sectional survey

**Jean Pierre Nganabashaka**[1]*, **Jean Berchmans Niyibizi**[1,2], **Ghislaine Umwali**[1], **Stephen Rulisa**[1], **Charlotte M. Bavuma**[1], **Jean Claude Byiringiro**[1], **Seleman Ntawuyirushintege**[1], **Pierre Celestin Niyomugabo**[1], **Lambert Izerimana**[1], **David Tumusiime**[1]

1 College of Medicine and Health Sciences, University of Rwanda, Kigali, Rwanda, 2 Global Public Health, Karolinska Institute, Stockholm, Sweden

* nganajp@gmail.com

**Data Availability Statement:** The 2012 Policy of health sector research in Rwanda requires that researchers desiring to access and use health

## Abstract

### Introduction

More than a third of the world's population was under full or partial lockdown during COVID-19 by April 2020. Such mitigation measures might have affected participation in various Physical activity (PA) and increased sedentary time. This study aimed to assess the effect of the COVID-19 mitigation measures on participation of adults in various PA types in Rwanda.

### Methods

We collected data from conveniently selected participants at their respective PA sites. We assessed the variation in time spent doing in four types of PA (Work related PA, PA in and around home, transportation PA and recreation, sport, and leisure purpose) across different pandemic period. We also evaluated the sedentary time over the weekdays and on the weekends.

### Results

A total of 1136 participants completed online assisted questionnaire. 71.4% were male, 83% of the study participants aged 18 to 35 years (mean = 29, (standard deviation = 7.79). Mean time spent doing vigorous PA as part of the work dropped from 84.5 minutes per day before COVID-19 to 58.6 minutes per day during lockdown and went back to 81.5 minutes per day after the lockdown. Time spent sitting on weekdays increased from 163 before COVID-19 to 244.5 minutes during lockdown and to 166.8 minutes after lockdown. Sitting time on weekend increased from 150 before COVID-19 to 235 minutes during lockdown and to 151 minutes after lockdown. Sleeping time on weekdays increased from 7.5 hours per day before COVID-19 to 9.9 hours during lockdown and to 7.5 hours after lockdown while it

research databases to consent to the country Ministry of Health, and comply with the "law of access to information" issued by the Rwanda Parliament, to access to research databases. Request for access to raw data used for this study findings should be addressed to the Rwanda National ethics community (RNEC) at info@rnecrwanda.org with a copy to the RNEC chairperson at jmazarati@gmail.com and to the corresponding author at nganajp@gmail.com/p.nganabashaka.ur.ac.rw.

**Funding:** This work was supported by the German Federal Ministry of Education and Research (Bundesministerium für Bildung und Forschung (BMBF)) (01KA1608) as part of the Research Networks for Health Innovation in Sub-Saharan Africa funding initiative. The funder had no role in designing and conducting the study or in writing the manuscript.

**Competing interests:** The authors have declared that no competing interests exist.

increased from 8 hours before COVID-19 to 10 hours during lockdown and to 8 hours per day after lockdown during weekends.

## Conclusion

The study emphasizes the significance of diverse PA, including home-based programs, during pandemics like COVID-19. It suggests promoting PA types like work-related, transportation, and domestic works during lockdown and similar period.

## Background

The COVID-19 pandemic became a serious global health threat [1]. As of August 21st, 2021, there were 411 million cases and around 4.42 million deaths globally [2]. On February 14th, 2020, the first case was reported in Egypt, making it the first country in Africa to have a COVID-19 case. One month after (on 14th March 2020), the Ministry of Health confirmed the first case of the virus in Rwanda. As of August 21st, 2021, 82,215 cases were confirmed, and 1,005 deaths were recorded in Rwanda [2]. The Government of Rwanda employed various mitigation measures including restrictions on movements, physical meetings, gatherings, as well as the temporary closure of social and economic activities. In particular, officially commenced a total lockdown on 22nd March 2020 [3] and partially lifted it in phases beginning on 4th May 2020 [4]. In early 2021 due to the increase in COVID-19 cases, a lockdown was introduced in Kigali from 19th January to 7th February 2021. Additionally, because of the upsurge in new cases of the COVID-19 pandemic, 10 days of total lockdown was imposed in Kigali city and eight other districts starting from 17th to 26th July 2021 [5]; this has been extended with additional 5 days starting from 27th to 31st July 2021 [6].

As part of efforts to control the spread of the COVID-19 pandemic, countries globally implemented a range of restrictive measures. For instance, by early April 2020, over one-third of the world's population was subjected to full or partial lockdowns [7].

During the total lockdown, unnecessary movements, and visits outside of home were prohibited, people were required to stay at home, shops and markets were closed except for those selling food, medicines, hygiene and sanitation products and other essentials items. Public and private sector employees were requested to work from home except those providing essential services. Additionally, physical activity (PA) facilities were closed, both individual and mass outdoor PA were not permitted. Upon closure of the outdoor PA campaigns and related facilities, Rwanda Biomedical Center (RBC) recommended people to keep exercising at home while keeping safe from COVID-19 trying home-based exercises such as performing domestic works, dancing, playing with children, walking in the compounds and taking online physical exercise class to get professional guidance [8]. In collaboration with the Rwanda Non-Communicable Diseases Alliance (RNCDA), a civil society organization that unites 27 different groups, including those advocating for individuals with non-communicable diseases (NCDs), healthcare professionals, and youth-led organizations within the NCDs community, the Ministry of Health and the Ministry of Sports have introduced a recorded series of modified PA sessions. These sessions are being rebroadcast on both national and private television channels, serving as a valuable guidance for individuals looking to engage in home-based PA during lockdown. In addition, during lockdown, the government has intermittently allowed people to exercise within their respective villages' catchment area. The adapted PA program is meant to be an important approach to maintaining PA and its preventive effects as well as a useful

complementary tool to improve the physical and psychological well-being of diverse populations affected by the COVID-19-pandemic [9].

Though the employed mitigation measures might have contributed to the virus containment, they forced the stop or slow implementation of PA interventions, which are among the ongoing population-level interventions targeting non-communicable diseases (NCD) risk factors. For example, Friday PA [10], this is a special time of two hours for public servants to exercise every Friday from 3 to 5 pm was closed. Additionally, a bimonthly PA campaign known as car-free day [11] was initiated in city of Kigali in 2016 that was latter extended to other cities and towns in the country was similarly stopped during lockdown. Movement restrictions associated with lockdown among other mitigation measures have exaggerated physical inactivity among types 2 diabetes mellitus patients [12], and compromised physical activity levels generally [13–16], and increased sedentary behavior [17]. Physical inactivity and sedentary behavior are linked to increased cardiovascular risks as well as other NCDs [18–20]. In the current pandemic, people suffering from NCDs are at higher risk of severe illness or death [21,22]. Further, reduction in PA have been proven to be linked with mood disorders [23]. To our knowledge, no study has documented this effect of COVID-19 on physical activity participation as a population-level intervention targeting risk factors for NCDs in Rwanda. This is a part of a large study aimed to evaluate the effect of COVID-19 pandemic mitigation measures on PA participation among adults in Rwanda. This paper focuses on the change on time spent in various types of PA and sedentary lifestyle across different COVID-19 phases (before emergence of COVID-19, during total lockdown, and when the total lockdown was lifted). Knowing that there were intermittent lockdowns in some selected areas of the country because of a rapid increase in COVID-19 cases, this study referred to the one occurred from 22nd March to 4th May 2022 because it has equally affected the whole country. This study aimed to conduct a comprehensive investigation of PA across different life domains. It emphasized the impact of COVID-19 on previously overlooked areas of PA, including those related to work, home, transportation, and recreational activities, which were found to be essential during challenging periods like lockdowns. This study findings provide invaluable insights for policymakers, offering a cornerstone for developing effective PA interventions during pandemics like COVID-19. This study's relevance is indeed beyond the current pandemic crisis, as its applicability remains relevant in similar future scenarios, encompassing both prevention and mitigation efforts.

## Methods

### Study design, participants and data collection procedure

This was a cross-sectional retrospective study. We collected data for PA participation across three periods: before emergence of COVID-19: February 2020, during total lockdown: 22nd March to 4th May 2020, and when the total lockdown was lifted: after 4th May 2020 in order to track the changes in PA participation and sedentary time. However, due to the COVID-19 pandemic situation, the outdoor PA were resumed with only individual PA possible, and it was difficult to determine the number of people attending PA at a given site. Participants were conveniently recruited from purposively selected PA sites both in Kigali and in other four selected districts in each province (Nyagatare in Eastern, Huye in Southern, Musanze in Northern and Rubavu district in Western Province) of Rwanda known to have regular PA campaigns operational before emergence of COVID-19 pandemic. To avoid the spread of COVID-19 infection, we opted to use online survey. Only adults (18 years old and above) from the selected site who have access to or could access smartphone or computer with internet connection were targeted for this study. The sample size was determined by priori power

analysis. It was calculated using G*Power 3.1.9.7 calculations. For one-way analysis of variance (F test) with small effect size (f2 = 0.10), α error probability = 0.05, with 80% power, the minimum sample size for this study was determined to be 1096.

Trained data collectors were allocated at each selected study site and conveniently recruited participants through a recruitment form (this was a designed form that had contact details of the participant as a way to follow up for any assistance in filling questionnaire to increase the response rate and data quality as well) until the required sample was reached. Participants were given contacts of the data collector whom they could refer to for any clarification or assistance. The data collectors approached participants from different physical activity sites (mainly at the start and arrival point for walkers, runners or joggers), in gymnasium, sportsgrounds. Additionally, snowballing approach was used to reach out to more participants, wherein reached participants especially coaches were asked to share the study information together with an online survey link with others in their network such as through sport clubs and WhatsApp groups.

We used the long form of international Physical Activity Questionnaire (IPAQ) to measure and analyze the Physical habitual changes mainly the PAes and sedentary time [24] after being adapted to Rwandan context. IPAQ was chosen for this study as it provides details of different PA types including those that are done at home through different domestic works. After agreeing to participate in the study, participants were shared with the questionnaire link (using Kobo Collect online survey) on their preferred platform (through WhatsApp or email) and were followed up by the trained data collectors for needed assistance in filling the questionnaire and any query. We collected four different types of physical activity (work related, recreation, sport, and leisure-time, transportation and PA done in and around home) as well as the sedentary time.

We assessed work related PA by assessing the time (minutes) spent per day doing usual work such as paid jobs, farming, volunteer work, course work and any other unpaid work that is outside of participant's home. PA in and around home was assessed by asking the time (minutes) spent per day doing various activities in and around home such as housework, gardening, general maintenance work and caring for the family. For transportation PA, a participant gave the number of minutes spent per day travelling (walking or riding) from place to place, including to places like work, market, pharmacies, and so on while recreation, sport, and leisure-time PA were assessed by asking the time spent per day doing PA solely for recreation, sport, exercise or leisure purpose. The sedentary time was examined by assessing the time spent per day sitting (while at work, at home, doing course work and during leisure time) and during sleeping. This included time spent sitting at a desk, visiting friends, reading or sitting or lying down to watch television. PA was categorized either vigorous if involved activities that highly increased breathing rate and or heart rate or moderate if it involved activities that slightly increased the breathing rate and or heart rate [25]. Data collection lasted for two months starting from May to June 2021.

## Data management and analysis

The statistician reviewed every submitted form to assess data submission progress, completeness, validity, and accuracy. Daily feedback was given to data collectors to address any raised concerns. Duplicate entries were removed prior to data analysis. The collected data were analyzed using STATA 16 [26]. Descriptive statistics such as mean, median, range and 95% confidence interval were used to describe demographics, PA time and sedentary time. For categorical variable types, frequencies and percentages were calculated. One-way analysis of variances was performed to examine the significant change in PA participation across the

three phases of the pandemic. The significant level was set at 0.05 with 95% confidence interval. The long form of international Physical Activity Questionnaire (IPAQ) was used to categorize and analyzed the sedentary lifestyle. This study followed STROBE guideline for cross-sectional studies reporting [27].

## Ethical considerations

All methods were performed in accordance with the ethical standards of the national research ethics committee and with the 1964 Helsinki declaration and its latter amendments [28]. Ethical approval was obtained from Rwanda National Ethics Committee (approval notice No. 856/RNEC/2021). The online survey started with an informed consent statement that provided full description of the study to enable a participant to take an informed and voluntary decision to participate in the study. After reading the informed consent statement, the participant was requested to select "agree" button to access the questionnaire if he/she decided to participate in the study or "disagree" button to end the form if he/she did not want to participate. Participants' decision (to agree or disagree to participate in the study) was recorded and retrieved from the server for archiving and witness purpose.

## Results

### Demographics of participants

A total of 1136 people were contacted and given questionnaire link, of which 23 voluntary rejected to participate and 1113 successfully completed the questionnaires. As presented in Table 1, the majority of participants (71.4%). Most of the study participants (44.8%) aged 26 to 36 years. The mean age of the respondents was 29 years old (standard deviation (SD) = 7.79). More than 58% of the participants had university level while only 1% did not have formal education. Regarding household size, 63.4% of the participants live with 2 to 4 housemates and only 1.5% live with 10 or more persons in the same household.

Regarding the employment status of the participants, this study revealed tremendous changes across three phases of the pandemic. There has been a decrease in number of employed people since the emerging of COVID-19 from 61.5% before to 29%. The latter has then increased a little bit to 53% after the total lockdown but still lower than before the pandemic.

### Changes in physical activity participation before, during and after COVID-19 period

This study revealed a significant change in time spent doing different PA across three periods of COVID-19 (before emergence of COVID-19 emergence, during lockdowns and after the lockdown was lifted). As shown in Table 2, this study generally found a significant decline in time spent in different PA types throughout three periods of COVID-19.

### Work related physical activity

The average time spent doing vigorous PA dropped from 84.5 minutes before emergence of COVID-19 to 58.6 minutes per day during lockdown and increased up to 81.5 minutes per day after total lockdown (p-value = 0.019). The Bonferroni test showed that the only significance changes was remarked during total lockdown comparing to before emergence of COVID-19 (p value = 0.019). Daily time spent doing moderate PA as part of work slightly declined from 65.6 minutes a day before emergence of COVID-19 to 56.2 minutes a day during the total lockdown, and slightly increased to 67.7 minutes per day after lockdown.

**Table 1. Characteristics of the study participants.**

| Variables | Frequency | Percentage |
|---|---|---|
| **Gender** | | |
| Male | 795 | 71.43 |
| Female | 318 | 28.57 |
| **Age Category** | | |
| 18–25 | 407 | 38 |
| 26–35 | 480 | 44.80 |
| Above 36 | 183 | 17.10 |
| **Highest education attained** | | |
| No formal education | 12 | 1.08 |
| Primary | 94 | 8.45 |
| Secondary | 308 | 27.67 |
| Technical and Vocational Education and Training (TVET) | 45 | 4.04 |
| University | 654 | 58.76 |
| **Size of the household** | | |
| 1 person | 99 | 8.91 |
| 2–5 persons | 704 | 63.37 |
| 6–10 persons | 291 | 26.19 |
| Above 10 persons | 17 | 1.53 |
| **Employment status before COVID-19** | | |
| Not employed | 429 | 38.54 |
| Employed | 684 | 61.46 |
| **Employment status during the total lockdown** | | |
| Not employed | 790 | 70.98 |
| Employed | 323 | 29.02 |
| **Employment status after the total lockdown** | | |
| Not employed | 523 | 46.99 |
| Employed | 590 | 53.01 |

**Table 2. Change in time (minutes) spent per day doing PA before, during and after COVID-19 period.**

| Type of physical activity (PA) | Before emergence of COVID-19 | During total Lockdown | After total lockdown | P value |
|---|---|---|---|---|
| **Work related physical activity** | | | | |
| Vigorous PA as part of work | 84.5 | 58.6 | 81.5 | 0.01* |
| Moderate PA as part of work | 65.6 | 56.2 | 67.7 | 0.31 |
| Walking PA as part of work | 63.2 | 65.2 | 63.2 | 0.97 |
| **Physical activity in or around home** | | | | |
| Vigorous PA in or around home | 55.6 | 54.3 | 53.8 | 0.92 |
| Moderate PA in or around home | 44.4 | 50.4 | 47.6 | 0.09 |
| **Transportation physical activity** | | | | |
| Walking PA from place to place | 49.5 | 37.3 | 47.9 | 0.002* |
| Riding bicycle from place to place | 54.9 | 49.5 | 50.5 | 0.61 |
| **Recreation, sport, and leisure type of physical activity** | | | | |
| Vigorous PA | 49.4 | 42.5 | 52.3 | 0.05* |
| Moderate PA | 38.6 | 42.9 | 46.9 | 0.06 |
| Walking PA | 46.9 | 37.3 | 47.8 | 0.03* |

However, this change was not statistically significant (p-value = 0.319). There was non-significant change in walking as part of the work, which varied from 63.2 minutes per day before emergence of COVID-19 to 65.2 minutes per day during the total lockdown and got back to 63.2 minutes per day after the lockdown.

### Physical activity in and around home

The participation in vigorous PA in and around home declined from 55.6 to 54.3 minutes per day during lockdown and slightly dropped to 53.8 minutes per day after the total lockdown (p-value = 0.924). The average time spent in moderate PA around home increased from 44.4 to 50.4 minutes per day during the lockdown and declined to 47.6 minutes per day after lockdown (p-value = 0.069). The change in both vigorous and moderate PA done in and around home across different phases of COVID-19 was not significant.

### Transportation physical activity

Walking from one place to another as well as bicycle riding are also recognized as types of PA. The respondents were asked the time spent travelling from place to place, either by foot (walking) or using the bicycle. The time spent walking from place to place declined from 49.5 minutes per day before emergence of COVID-19 to 37.3 minutes per day during the total lockdown and increased to 47.9 minutes per day after total lockdown (p-value = 0.002). The Bonferroni test showed that the significant change was identified during total lockdown comparing to before emergence of COVID-19 (p-value = 0.002) and after COVID-19 (p-value = 0.01) but no significant changes found between before and after COVID-19 (p-value = 1).

Considering the time spent riding bicycle, there was non-significant changes across three COVID-19 phases. The time dropped from 54.9 minutes per day before emergence of COVID-19 to 49.5 minutes per day during total lockdown and reached 50.5 minutes per day after total lockdown (p-value = 0.61).

### Physical activity for recreation, sport, and leisure purpose

Apart from PA related to work, domestic works and transportation, this study assessed the changes in PA done solely for recreation, sports or leisure purpose. The average minutes per day spent walking for leisure purpose declined from 47 minutes per day before emergence of COVID-19 to 37 minutes per day during the total lockdown and rose to 48 minutes per day after total lockdown (p-value = 0.03). Both vigorous and moderate recreational, sports or leisure type of PA were reduced upon the emergence of COVID-19 (during total lockdown) and slightly increased when the movements' restriction measures were lifted (after the total lockdown); however, only change to vigorous PA was significant (p-value 0.05). The Bonferroni test showed that a significant change in time spent in PA for recreation, sport, and leisure purpose was found during total lockdown comparing to both before **emergence of** COVID-19 (p-value = 0.03) and after total lockdown (p-value = 0.03).

### Sedentary time (Sitting and sleeping time)

As shown in **Table 3**, the average of daily sitting time on a workday sharply increased from 163.5 minutes (approximately 2 hours 44 minutes) per workday before the emergence of COVID-19 to 244.5 minutes (around 4 hours) per workday during the lockdown and get back to 166.8 minutes per day when the lockdown was lifted. The sitting time increased from 150 minutes per weekend day before emergence of COVID-19 to 235.8 minutes per weekend day

**Table 3. Daily average sedentary time per day throughout three phases of COVID-19.**

| Time | Before emergence of COVID-19 | During total lockdown | After total lockdown | P value |
|---|---|---|---|---|
| **Sitting time (Minutes)** | | | | |
| On a workday | 163.5 | 244.5 | 166.8 | 0.000* |
| On a weekend day | 150.0 | 235.2 | 151.0 | 0.000* |
| **Sleeping time (hours)** | | | | |
| On a workday | 7.5 | 9.9 | 7.5 | 0.000* |
| On a weekend day | 8.0 | 10.3 | 8.1 | 0.000* |

during the total lockdown and has declined to 151 minutes per day after the total lockdown. These changes in both sitting and sleeping time was statistically significant (p-value < 0.001).

The average sleeping time including naps on workday increased from 7 hours and half a day before emergence of COVID-19 to 10 hours during the lockdown and declined back to 7 hours once the lockdown was lifted (p-value < 0.001). The same trend line was found on weekend days. The sleeping time increased from 8 hours a day before emergence of COVID-19 to 10 hours a day on weekend day during the lockdown and declined back to 8 hours a day after the lockdown (p- value < 0.001).

The Bonferroni test showed that a significant increase in sitting time on weekdays was found during total lockdown comparing to both before emergence of COVID-19 (p-value = 0.036) and after total lockdown (p-value = 0.030). The same increase in sitting time was identified on weekends during lockdown compared to before emergence of COVID-19 (p-value = 0.001) and after total lockdown (p-value = 0.001). There was a significant increase in sleeping time during total lockdown both on weekdays and weekends comparing both before emergence of COVID-19 (p-value = 0.001) and after total lockdown (p-value = 0.001).

## Discussion

In addition to causing immediate illnesses and fatalities, COVID-19 also raised various health hazards that could have an impact on people's health. This study's distinctiveness lies in its comprehensive exploration of PA across diverse life domains, shedding light on previously overlooked areas that proved essential during challenging times like lockdowns, while also offering insights into the pandemic's effects on sedentary behavior and advocating for a broader approach to promoting various forms of PA.

The majority (83%) of the study participants were aged less or equal to 35 years old and males (71.4%) with university level literate (58%). According to the national population census [29], Rwanda is dominated by young people where 62% of the population are under 25 years, this might explain the youth dominance in this study participation. In addition, the fact that participants were recruited from PA sites where people are doing various aerobic exercises and playing different games from which, you might be more involved than adults may explain the dominance of youth in this study. The old people are likely to exercise in nearby places [30] and sometimes present some barriers and reluctance to PA [31], which might explain the youth dominance of this study participants. Females were less represented in this study compared to males, which shows a gap in PA practice among both gender; this confirmed what was found in studies conducted in Ireland [32] and Australia [33]. The evidence showed that females are less involved in PA than Educated people are likely to have knowledge of PA benefits than the less literate and studies have linked PA level with education [34], though we did not intent to measure the association of education and PA. Various studies have evidenced a

positive relationship [35–37] which might explain the magnitude of participants educated at university level.

Generally, COVID-19 mitigation measures have significantly reduced time spent in PA and increased sedentary lifestyle (sitting and sleeping) time across different period of COVID-19. The significant reduction in different PA types was found in lockdown compared to the before emergence of COVID-19 period. A slight increase was noticed after total lockdown when the mitigation measures were lifted. This finding confirms what were found in other studies [38–41].

This study revealed a significant decrease of work-related PAs; this reduction was reported during lockdown particularly. During total lockdown, many businesses were closed. This closure resulted in people losing job and reduce their movement from their home to their workplace and vice-versa. Especially, our sample was recruited from the cities where during total lockdown, most of the activities (businesses, schools, etc) were closed apart from the essential services. Additionally, the majority (61%) of the participants reported to be employed (paid or unpaid work) outside of their home, therefore the closure of services (businesses) judged as not essential may explain the significant decrease of the PA as part of the work. Similarly, a review conducted in 2021 found that COVID-19 mitigation measures such as lockdowns has compromised PAs [42]. While WHO recognizes all movement including during leisure time, transportation to get to and from places, or as part of a person's work as PA which equally improve people' health [43]. There are some controversial arguments that do not positively associate work related PA with health benefits [44] and tend to promote only leisure time PA [45].

Participation in PA in and around home did not significantly change across three periods. This was not a surprising finding since lockdown and other COVID-19 mitigation measures did not restrict PA done in and around home, rather the latter were highly encouraged [8] because people were expected to stay at home especially during total lockdown; however, we noted that the mean time spent in this type of PA did not significantly change. Although the literature indicates that performing housework only cannot provide the recommended PA time [46], the household activities offer the same benefits as other PA and significantly contribute to the achievement of the recommended level of PA [47]. These findings contrast with what was found in German population [48], this might be due to the sample size as well as the different context. In Rwandan context, housework is mostly done by hired housemaids, children and women men are rarely engaged in house works. This study showed that PA done in terms of transportation (walking and riding from place to place) was significantly dropped. Especially during lockdown, only essential services were operating thus only people seeking such services were allowed to move outside of their home with condition to minimize number of times and the duration spent in these activities to reduce the risk of getting COVID-19 infection, which might have led to reduced time spent in this type of PA. During lockdown, most employees (public and private sector) were advised to work from home, which reduced the time spent in active travel (walking and riding) from home to work and the other way around. This finding confirm what was found in a study conducted among German Population [48]. This finding however contrasts with some countries, where people were permitted to go out for exercise during lockdown, which promoted individual active travel (walking, bicycle riding) and increase time spent in this kind of PA [49].

During the lockdown, the amount of time people spent on recreational activities, sports, and leisure PA significantly decreased. Rwanda had previously promoted outdoor and group PA initiatives, like car-free days in cities [11] and nationwide mass PA campaigns. However, indoor, individual, and home-based PA were not widely encouraged or popular among the general population, possibly because they seemed challenging and less enjoyable, especially for those accustomed to outdoor PA with friends.

The reduction in this types of PA was similarly found in other studies around the world [50–52]. Like other types of PA, there were a slight increase in recreational PA type after

lockdown. Studies have revealed that people with self-efficacy for PA have a strong likelihood to be physically active [53] and a great ambitious to have a long term PA commitments [54]; this validate that people who were used to outdoor PA were likely to resume as far as they are permitted to do so. Moreover, the resumption of PA facilities such as gymnasium, playgrounds and roads sidewalks with strict respect to COVID-19 prevention measures such hygiene and social distancing measures allowed people to get back in their PA routine.

Sedentary time (sitting and sleeping time), increased significantly during total lockdown and returned to normal after total lockdown. As time spent in PA reduced, there was an increase in time spent in sitting and sleeping on both weekends and weekdays. When the total lockdown was imposed, many people lose their occupations, no more unnecessary walk around from place to place, no visit and any other optional movement were permitted, the only options people were left with was to stay at their home. People might have used this time homes watching TV screens, playing cards and other games on and offline games and sleeping, which might have contributed to the increase of their sitting and sleeping time. This finding confirm what were shown in other studies, for example a study done among students from 2 Canadian universities showed that they were not interested in PA which increased their sedentary behaviour [55]; other studies around revealed the similar findings [56,57]. It is important to note that each reduction regardless of the extent in PA and increase in sedentary lifestyle increase the risk of various NCDs [58]. This study was a cross-sectional type in nature and relied on retrospective data collected through online survey, because of this nature, sampling bias was unescapable and generalizability is restricted. Again, this study used self-reported data on which desirability and sampling bias was not sufficiently controlled. However, the repetitive data collection at different site might have increased the participants' diversity, which makes our findings robust. The cultural, religious and other social-demographic influences should be considered in this kind of studies in futures and future studies to explore how best PA can be improved during difficult time such as lockdown are recommended.

## Conclusion

This study investigated PA across various life domains, including transportation, recreation, and household chores, some of which were previously underexplored but proved valuable during challenging periods like lockdowns. It examined the impact of the pandemic on people's sedentary behavior. It provides valuable reminder for intervention planners and policymakers to expand their focus beyond outdoor recreational activity and promote other forms of physical activity, including home-based, indoor, and work-related activities. Home-based PA programs are efficient, effective and beneficial during pandemic such as COVID-19. People should be mobilised to get used to home-based and indoor PA. Furthermore, there is a need of strategies encouraging people to be engaged in other types of PAs such as work-related PA, transportation and domestic works that are believed to have similar benefits with recreational, sports and leisure type of PA. Indeed, the latter do not require additional budget and infrastructures since there are done where people live and work without a need of more equipment.

## Acknowledgments

We would like to express our gratitude to the participants in this study and acknowledge significant contributions of Aline Ikirezi and Kerstin Sell to the success of this work.

## Author Contributions

**Conceptualization:** Jean Pierre Nganabashaka, Stephen Rulisa, Seleman Ntawuyirushintege, Pierre Celestin Niyomugabo, David Tumusiime.

**Data curation:** Jean Pierre Nganabashaka, Jean Berchmans Niyibizi, Lambert Izerimana.

**Formal analysis:** Jean Pierre Nganabashaka, Pierre Celestin Niyomugabo, Lambert Izerimana.

**Investigation:** Stephen Rulisa, Charlotte M. Bavuma, David Tumusiime.

**Methodology:** Jean Pierre Nganabashaka, Jean Berchmans Niyibizi, Ghislaine Umwali, Stephen Rulisa, Charlotte M. Bavuma, Jean Claude Byiringiro, Seleman Ntawuyirushintege, Pierre Celestin Niyomugabo, Lambert Izerimana, David Tumusiime.

**Project administration:** Seleman Ntawuyirushintege.

**Supervision:** Jean Pierre Nganabashaka, Ghislaine Umwali, Pierre Celestin Niyomugabo, Lambert Izerimana, David Tumusiime.

**Validation:** Charlotte M. Bavuma, Jean Claude Byiringiro, Seleman Ntawuyirushintege, David Tumusiime.

**Writing – original draft:** Jean Pierre Nganabashaka.

**Writing – review & editing:** Jean Berchmans Niyibizi, Ghislaine Umwali, Stephen Rulisa, Charlotte M. Bavuma, Jean Claude Byiringiro, Seleman Ntawuyirushintege, Pierre Celestin Niyomugabo, Lambert Izerimana, David Tumusiime.

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
