## [Decision Letter · Decision Letter 0]

10 Apr 2023

PONE-D-22-31575The effects of Covid-19 mitigation measures on physical activity participation among adults in RwandaPLOS ONE

Dear Dr. NGANABASHAKA

Thank you for submitting your manuscript to PLOS ONE. After careful consideration, we feel that it has merit but does not fully meet PLOS ONE’s publication criteria as it currently stands. Therefore, we invite you to submit a revised version of the manuscript that addresses the points raised during the review process.

ACADEMIC EDITOR: Please revise the manuscript based on the comments raised and also align the manuscript to PLOS ONE's requirements.

Please submit your revised manuscript 8th May 2023. If you will need more time than this to complete your revisions, please reply to this message or contact the journal office at plosone@plos.org. Please include the following items when submitting your revised manuscript:A rebuttal letter that responds to each point raised by the academic editor and reviewer(s). You should upload this letter as a separate file labeled 'Response to Reviewers'.A marked-up copy of your manuscript that highlights changes made to the original version. You should upload this as a separate file labeled 'Revised Manuscript with Track Changes'.An unmarked version of your revised paper without tracked changes. You should upload this as a separate file labeled 'Manuscript'.If applicable, we recommend that you deposit your laboratory protocols in protocols.io to enhance the reproducibility of your results. Protocols.io assigns your protocol its own identifier (DOI) so that it can be cited independently in the future. For instructions see: https://journals.plos.org/plosone/s/submission-guidelines#loc-laboratory-protocols. Additionally, PLOS ONE offers an option for publishing peer-reviewed Lab Protocol articles, which describe protocols hosted on protocols.io. Read more information on sharing protocols at https://plos.org/protocols?utm_medium=editorial-email&utm_source=authorletters&utm_campaign=protocols.

We look forward to receiving your revised manuscript.

Kind regards,

Mpho Keetile, PhD

Academic Editor

PLOS ONE

Additional Editor Comments:

The reviewer has recommended that you address some minor comments for the article. I also recommend the same. Please make sure that your article adhere to all the journal requirements

Reviewers' comments:

Reviewer's Responses to Questions

**Comments to the Author**

1. Is the manuscript technically sound, and do the data support the conclusions?

Reviewer #1: Yes

2. Has the statistical analysis been performed appropriately and rigorously? 

Reviewer #1: Yes

3. Have the authors made all data underlying the findings in their manuscript fully available?

Reviewer #1: Yes

4. Is the manuscript presented in an intelligible fashion and written in standard English?

Reviewer #1: Yes

5. Review Comments to the Author

Reviewer #1: 1- I suggest including in the title some information about the type of study (a cross-sectional online survey on COVID-19 or a cross-sectional retrospective study.).

2- Abstract- Methods: What was the target population for this study? How was the questionnaire disclosed and accessed by the participants? The IPAQ could also be mentioned here as an instrument for the analysis of sedentary lifestyle.

3- line 100: Please, withdraw 1 “study” (This was a cross-sectional retrospective study.)

4- line 112: It was calculated using G*Power 3.1.9.7 calculations. Using G*Power 3.1.9.7 (is repetitive).

5- line 127-128: This sentence deserves one more reference today.

6- line 152-154: The effect of COVID-19 related mitigations measures on physical activity were measured through the changes in duration spent in different physical activities. (this content shouldn't be here)

7- This current study must have followed some guideline (maybe STROBE?) https://www.strobe-statement.org/checklists/. It would be very important to include this information.

8- Declaration of Helsinki needs to be cited in ethical considerations as well.

9- Tabela 2: include in the legend, the description for the acronym PA and inform the level of significance considered in these analysis.

10- line 153, 154, 199, 202, 261, 264, 267, 268, 269, 284, 287, 290, 295, 296, 312, 313, 320, 327, 342, : use PA for physical activity.

11- Although the authors consider that data collected in different locations with a diverse sample, make the results more robust; I think it is worth noting here the cultural and religious differences that may justify certain findings.

6. PLOS authors have the option to publish the peer review history of their article (what does this mean?). If published, this will include your full peer review and any attached files.

Reviewer #1: **Yes: **Laisa Liane Paineiras-Domingos

---

## [Author Response · Author response to Decision Letter 0]

6 May 2023

The Resonse letter with all comments raised by the reviewiers addressed was submitted in the system.

---

## [Decision Letter · Decision Letter 1]

1 Aug 2023

PONE-D-22-31575R1The Effects of COVID-19 Mitigation Measures on Physical Activity Participation among Adults in Rwanda: an Online Cross-sectional SurveyPLOS ONE

Dear Dr. NGANABASHAKA,

Thank you for submitting your manuscript to PLOS ONE. After careful consideration, we feel that it has merit but does not fully meet PLOS ONE’s publication criteria as it currently stands. Therefore, we invite you to submit a revised version of the manuscript that addresses the points raised during the review process.

ACADEMIC EDITOR: 

Please address the minor comments shared and your article will be ready for acceptance to publication. 

We look forward to receiving your revised manuscript.

Kind regards,

Mpho Keetile, PhD

Academic Editor

PLOS ONE

Comments from PLOS Editorial Office:

We note that one or more reviewers has recommended that you cite specific previously published works. As always, we recommend that you please review and evaluate the requested works to determine whether they are relevant and should be cited. It is not a requirement to cite these works. We appreciate your attention to this request.

Journal Requirements:

Additional Editor Comments:

Please address the reviewers comments and your article will be recommended for publication. If you address the comments well the article may be publishable

Reviewers' comments:

Reviewer's Responses to Questions

**Comments to the Author**

1. If the authors have adequately addressed your comments raised in a previous round of review and you feel that this manuscript is now acceptable for publication, you may indicate that here to bypass the “Comments to the Author” section, enter your conflict of interest statement in the “Confidential to Editor” section, and submit your "Accept" recommendation.

Reviewer #2: All comments have been addressed

Reviewer #3: (No Response)

Reviewer #4: (No Response)

2. Is the manuscript technically sound, and do the data support the conclusions?

Reviewer #2: No

Reviewer #3: Partly

Reviewer #4: Partly

3. Has the statistical analysis been performed appropriately and rigorously? 

Reviewer #2: I Don't Know

Reviewer #3: Yes

Reviewer #4: I Don't Know

4. Have the authors made all data underlying the findings in their manuscript fully available?

Reviewer #2: Yes

Reviewer #3: Yes

Reviewer #4: Yes

5. Is the manuscript presented in an intelligible fashion and written in standard English?

Reviewer #2: Yes

Reviewer #3: No

Reviewer #4: Yes

6. Review Comments to the Author

Reviewer #2: work more on abstract. here, you can see recommendations:

Overall, the abstract provides a clear and concise summary of the study's objectives, methods, and results. However, there are some areas where the abstract could be improved:

The abstract does not clearly state the type of physical activity that was assessed. The study evaluated the variation in time spent doing different PA types, but the abstract does not provide any information on what those types were.

The abstract could benefit from more specific information on the study population. While the abstract mentions that 1136 adults above 18 years old completed the questionnaire, it does not provide any information on their demographic characteristics or how they were selected.

it may be useful to highlight the importance of this research, even four years after the emergence of COVID-19. While the study's findings provide valuable insights into how COVID-19 mitigation measures impact physical activity participation in Rwanda, the implications of these findings extend beyond the current pandemic. use thefollowin references for that:

Taheri, Morteza, et al. "Effects of home confinement on physical activity, nutrition, and sleep quality during the COVID-19 outbreak in amateur and elite athletes." Frontiers in nutrition 10 (2023): 1143340.

Reviewer #3: Overall this is an interesting study which could have much interest, however, I do have some concerns as follows:-

In the list of authors, you have listed No 2 Pettenfer School of Public health, but none of the authors are cited at that institution. Please clarify

You have stated a very low number of Covid cases in Rawanda, which is due to low amounts of testing. It would be good to give this detail as many readers will not realise why the number of cases are so low. This has been supported by the World Health Organisation who states that the numbers are low due to low testing.

In the abstract, it would be good to mention that statistical analysis was carried out rather than just listing percetage change.

Line 65 upon not up on

Line 69 what is NCD alliance?

Line 70 rebroadcast not rebroadcasted

Line 71 should be 'activity program is believed' not 'are believed'

Line 72 mainaining not maintain

Line 146 – serious reservations about this time period as many counties were still in and out of lockdown, therefore had people gone back to pre covid siuation? Were government backed PA activities back up and running? if not this will scew your results as people were not completed out of lockdown.

Line 167 was 29 years not with 29 years

Line 166 put 71.4% in brackets

Is it a bias that 58.76% have university education when statistics show that in 2019 only 12.8% of Rawandan population had higher education?

Line 175 should be latter not later

Line 191 what is meant by 'walking as part of the work'? This seems to have slightly increased during total lockdown when hardly anyone was at work. This needs explaining.

Line 212 again it doesn’t make sense how bike riding only reduced by 5 mintes a day if you were locked down

What is the difference between daily sitting time and sedentary sitting time line 228? Please explain

Line 240 there is a repeat of half a sentence about the Bonferroni test, please remove

Paragraph starting line 257 is hard to understand and the whole paragraph needs re-wording.

Line 263 explain why old people present barriers to PA, especially when you are talking of people age 35y and over, this isn’t particularly old

Line 274 shoud be before the emergence of covid 19 not the before

Line 275 where were these other studies that are referred to ? in Rawanda or other countries?

Line 308 ; other countries allowed to go outside and exercise during lockdown, but not Rawanda, so how did they bike ride duing lockdown? You have said that this only decreased by 5% during lockdown, which doesnt make sense.

Line 323 not resume but resumption

Line 341 this section doesn’t make sense. You have said that sitting time increased in your results but now say that it did not increase during lockdown.

Needs more discussion on limitations of the study eg that is was retrospective and the bias of males vs females and age

Does the conclusion contradict earlier statements about home based PA programmes as you are saying they are efficient but earlier you have said that people get bored of them?

Some of the english is difficult to understand and I would suggest that you get someone with english as a first language to proof read this.

Reviewer #4: Thank you for your submitted manuscript entitled, PONE-D-23-13290 ‘’The Effects of COVID-19 Mitigation Measures on Physical Activity Participation among Adults in Rwanda: an Online Cross-sectional Survey’’.

ABSTRACT:

• In the Methods provides more information about the PA and sedentary time assessment used in the study.

• How you determined the ‘’self-administered questionnaire’’.

• Could be a relevant conclusion for the present study. However, you note in the conclusion that ‘’COVID-mitigation measures have significantly reduced PA participation and increased sedentary time’’.. This statement is not a new discovery. Although, what is the new discovery?

INTRODUCTION:

• Line 97-108: The development of the introduction needs to be more hypotheses driven and develop the questions leading up to the section in the methods section. In this section, there is no new information in the paper this is reported in many published papers, it is necessary to remake and reinforce this part by other evidence, information and bibliographic references. Further, the part on hypothesis/-es development needs revision. A more conclusive use of appropriate literature can help here to clearly state the purpose of the study in order to develop hypotheses (or null hypotheses, whichever way is preferred). In turn, this will serve as the paper's framework as it will define the importance of findings, i.e. which results are being presented. In its present form, the paper states one quite unspecific hypothesis, therefore, results presented seem random which also influences the discussion.

METHOD

• The procedure is rigorous and well described and the statistical analysis is correct.

RESULTS:

• What about the measure of data reliability in the study?

• The tables are poorly presented, need formatting. Take time to check all formatting and make sure that all of the tables you have are mentioned in the text of the paper so that the reader can find them.

DISCUSSION:

• Line 275-279: The authors still need to clarify why this was done, and why it is important. This must be done in the Introduction, and clarified again here.

• Line 280-281: However, at no point is it clear why this information is important to the reader. The authors do mention conceptual reasons for the study in the introduction and sparsely in other sections, but this needs to be consistently done throughout the manuscript.

CONCLUSION

• Finally, improve the conclusion section: it is important to suggest possible future studies.

7. PLOS authors have the option to publish the peer review history of their article (what does this mean?). If published, this will include your full peer review and any attached files.

Reviewer #2: **Yes: **Morteza Taheri, Full Professor, university of Tehran

Reviewer #3: No

Reviewer #4: **Yes: **Souhail Hermassi

---

## [Author Response · Author response to Decision Letter 1]

22 Sep 2023

all the comments are addressed in the Response to reviewers

---

## [Editor Report · Decision Letter 2]

10 Oct 2023

The Effects of COVID-19 Mitigation Measures on Physical Activity Participation among Adults in Rwanda: an Online Cross-sectional Survey

PONE-D-22-31575R2

Dear Dr. Jean Pierre NGANABASHAKA

We’re pleased to inform you that your manuscript has been judged scientifically suitable for publication and will be formally accepted for publication once it meets all outstanding technical requirements.

Kind regards,

Mpho Keetile, PhD

Academic Editor

PLOS ONE
---

## [Editor Report · Acceptance letter]

16 Oct 2023

PONE-D-22-31575R2 

The Effects of COVID-19 Mitigation Measures on Physical Activity (PA) Participation among Adults in Rwanda: an Online Cross-sectional Survey 

Dear Dr. Nganabashaka:

I'm pleased to inform you that your manuscript has been deemed suitable for publication in PLOS ONE. Congratulations! Your manuscript is now with our production department. 

Kind regards, 

on behalf of

Dr. Mpho Keetile 

Academic Editor

PLOS ONE